# Immunotherapy in Breast Cancer

**DOI:** 10.3390/ijms25147517

**Published:** 2024-07-09

**Authors:** Kathrin Dvir, Sara Giordano, Jose Pablo Leone

**Affiliations:** 1Dana Farber Cancer Institute, Boston, MA 02215, USA; kathrin.dvir@gmail.com (K.D.);; 2St. Elizabeth’s Medical Center, Boston, MA 02111, USA

**Keywords:** breast cancer, immunotherapy, checkpoint inhibitors, biomarkers

## Abstract

Breast cancer is a disease encompassing a spectrum of molecular subtypes and clinical presentations, each with distinct prognostic implications and treatment responses. Breast cancer has traditionally been considered an immunologically “cold” tumor, unresponsive to immunotherapy. However, clinical trials in recent years have found immunotherapy to be an efficacious therapeutic option for select patients. Breast cancer is categorized into different subtypes ranging from the most common positive hormone receptor (HR+), human epidermal growth factor receptor 2 (HER2)—negative type, to less frequent HER2− positive breast cancer and triple-negative breast cancer (TNBC), highlighting the necessity for tailored treatment strategies aimed at maximizing patient outcomes. Despite notable progress in early detection and new therapeutic modalities, breast cancer remains the second leading cause of cancer death in the USA. Moreover, in recent decades, breast cancer incidence rates have been increasing, especially in women younger than the age of 50. This has prompted the exploration of new therapeutic approaches to address this trend, offering new therapeutic prospects for breast cancer patients. Immunotherapy is a class of therapeutic agents that has revolutionized the treatment landscape of many cancers, namely melanoma, lung cancer, and gastroesophageal cancers, amongst others. Though belatedly, immunotherapy has entered the treatment armamentarium of breast cancer, with the approval of pembrolizumab in combination with chemotherapy in triple-negative breast cancer (TNBC) in the neoadjuvant and advanced settings, thereby paving the path for further research and integration of immune checkpoint inhibitors in other subtypes of breast cancer. Trials exploring various combination therapies to harness the power of immunotherapy in symbiosis with various chemotherapeutic agents are ongoing in hopes of improving response rates and prolonging survival for breast cancer patients. Biomarkers and precise patient selection for the utilization of immunotherapy remain cardinal and are currently under investigation, with some biomarkers showing promise, such as Program Death Lignat-1 (PDL-1) Combined Positive Score, Tumor Mutation Burden (TMB), and Tumor Infiltrating Lymphocytes (TILs). This review will present the current landscape of immunotherapy, particularly checkpoint inhibitors, in different types of breast cancer.

## 1. Introduction

Breast cancer continues to be one of the most common and challenging malignancies globally, presenting a significant public health concern [1]. It is associated with significant morbidity and mortality despite advancements in conventional therapies [2,3]. Although advancements in early-stage and metastatic settings have significantly improved outcomes, significant challenges remain. Specifically, there is a need to address tumor heterogeneity and varied responses to neoadjuvant therapies, improve the rates of Pathologic Complete Response (pCR), mitigate the risks of recurrence in high-risk early breast cancers, and increase the response-adjusted treatment strategies that would allow a personalized approach and de-escalation of therapy in select patients. In the advanced/metastatic setting, there is a great need to expand the existing arsenal of therapeutic options while minimizing toxicities and financial burdens and preserving quality of life. In recent years, immunotherapy has emerged as a promising avenue for cancer treatment, harnessing the body’s immune system to target and eradicate tumor cells. It has become the standard of care in many cancer types such as lung cancer, melanoma, and others as well as in select patients with TNBC.

International research efforts over the past decades have highlighted both the achievements and limitations of various immunotherapeutic approaches. Immune Checkpoint Inhibitors (ICIs) have been investigated in breast cancer as single agents and in combination with chemotherapy and targeted therapies such as monoclonal antibodies, antibody-drug conjugates, Cyclin-dependent kinase (CDK) 4/6 inhibitors, and more. Key trials in this space have demonstrated variable outcomes for ICIs, with some showing remarkable results leading to their registration and incorporation into clinical guidelines [4,5,6], while others revealed futility or safety concerns and were promptly discontinued. These mixed results suggest a need for a deeper understanding of the tumor microenvironment and its interplay with ICIs, both alone and in combination with other therapeutics. Consequently, research efforts have been directed toward identifying biomarkers to aid in careful patient selection and prognostication, sparing non-responders from immune-related adverse events.

In this review, we present pivotal trials and current data on immunotherapy in breast cancer, focusing primarily on ICIs that have set current standards as reflected in the NCCN Guideline Version 3.2024. This review aims to consolidate existing knowledge and present the evolving landscape of immunotherapy in breast cancer. We hope this work guides researchers and clinicians in improving patient outcomes through innovative immunotherapeutic strategies.

## 2. How Does Immunotherapy Work?

Immunotherapy drugs that are currently utilized in breast cancer comprise of Checkpoint Inhibitors (ICI), namely PD-1 and PDL-1 inhibitors and HER2-directed monoclonal antibodies [7]. ICI works by blocking checkpoint proteins from binding with their partner proteins. This prevents the “off” signal from being sent, allowing the T cells to kill cancer cells [8,9,10,11].

Cancer cells employ adaptive strategies to evade detection and destruction. One way it is achieved is by promoting the binding of PD-L1 to PD-1, which in turn prevents immune T cells from killing tumor cells. By blocking the binding of PD-L1 to PD-1, ICI allow immune T cells to recognize and kill tumor cells. Simply put, these drugs unleash the full potential of immune cells by releasing the brakes that hinder their anti-tumor activity, allowing body’s own immune system to recognize and destroy cancer cells, which were previously undetected or tolerated by the immune system. This mechanism of action is also responsible for the immune-related adverse effects (irAE) that are seen with ICI use, resulting in autoimmunity that can be severe, irreversible, and can rarely be fatal [8,9].

The rationale behind combining immunotherapy with chemotherapy lies in their collaborative effects. Several commonly utilized chemotherapeutic agents, including anthracyclines, cyclophosphamide, and taxanes, promote cell death, resulting in the release of tumor-associated antigens and danger signals that recruit antigen-presenting cells, promote engulfment of dying cells, which promotes dendritic cells and T-cell priming. Concurrently, checkpoint inhibitors unleash cytotoxic immune T cells, enhancing tumor cell killing. This dual mechanism potentiates antitumor efficacy, leading to improved response rates and prolonged survival [10,11].

## 3. Current Landscape of ICI in Breast Cancer

To date, the US FDA has approved three different categories of ICIs. PD-1 inhibitors (Nivolumab, Pembrolizumab, Cemiplimab, and Dostarlimab), PDL-1 inhibitors (Atezolizumab, Durvalumab, and Avelumab), and a CTLA-4 inhibitor (Ipilimumab) [7]. Currently, only Pembrolizumab has USA FDA approval in TNBC space, after Atezolizumab was retracted from the US market following Impassion131 data analysis [12]. Indeed, not all patients respond equally to these drugs, highlighting the need for careful patient selection for predicting responses to immunotherapy and expanding our immunotherapy arsenal. 

### 3.1. Triple-Negative Breast Cancer (TNBC)

TNBC is an aggressive subtype of breast cancer with inferior outcomes and limited treatment options [13]. It is characterized by the absence of estrogen receptor (ER), progesterone receptor (PR), and is human epidermal growth factor receptor 2 (HER2) non-amplified. The traditional endocrine therapies are therefore ineffective in TNBC, and chemotherapy was the only systemic option available in this disease subtype, until the advent of immunotherapy [13].

Despite lacking canonical targets for endocrine therapy, TNBC is known to be responsive to ICI. This has been hypothesized to be associated with its relatively high tumor mutational burden (TMB) compared to other subtypes of BC [14].

Indeed, checkpoint inhibition with pembrolizumab has been approved for early stage usage following a landmark KEYNOTE 522 trial [4] and in the advanced stage, PD-L1-positive TNBC has been approved based on the improvement in outcomes observed when combined with frontline chemotherapy in the KEYNOTE 355 trial [5]. Notably, evidence suggests that there is a superior efficacy of ICIs in TNBC when administered early in the disease course, possibly due to a non-compromised immune system due to cytotoxic therapies or the progression of immune escape mechanisms during the advancement of the disease [14,15,16].

#### 3.1.1. ICI in Early Stage TNBC

Around 60% of TNBC patients in the USA are diagnosed with stage II or III disease. Nonetheless, it carries a high rate of recurrence (3-year distant recurrence rate of 30–35%) and the poorest prognosis among all BC subtypes (5-year estimate OS of 64% for stages I–III combined), emphasizing the need to improve therapy in this setting [17,18,19]. Table 1 summarizes the pivotal trial in this space.

The new standard of care in high-risk stage II and III TNBC is the KEYNOTE 522 regimen that utilizes neoadjuvant paclitaxel–carboplatin with pembrolizumab, followed by doxorubicin–cyclophosphamide with pembrolizumab, as was established in a phase 3 RCT [4] study that evaluated pembrolizumab alongside standard chemotherapy in 1174 patients with high risk, early-stage TNBC. The study demonstrated improved pathologic complete response (pCR) rates with pembrolizumab; 65% vs. 51% with a placebo [4,15].

After a median follow-up of 63.1 months [6,21], neoadjuvant pembrolizumab plus chemotherapy followed by adjuvant pembrolizumab achieved a 5-year EFS of 81.3% versus 72.3% with neoadjuvant chemotherapy with a placebo, translating into a reduction in risk for recurrence, progression, complications, or death by 37% (HR, 0.63; 95% CI, 0.49–0.81). Additionally, the benefit in EFS was consistent across subgroups stratified by PD-L1 status, nodal status, tumor size, chemotherapy schedule, age, and ECOG performance status, and the median EFS was not achieved in either group. Although the improvement in EFS with pembrolizumab was seen regardless of pathologic complete response (pCR) outcomes, additional analysis showed that the highest rates of 5-year EFS were observed in patients who achieved pCR after neoadjuvant therapy, emphasizing the link between pCR and long-term outcomes in TNBC. 

Two other pivotal trials have demonstrated positive results when incorporating immune check inhibitors in the neoadjuvant setting of TNBC: the I-SPY2 and the Impassion031.

Similarly, The Impassion031 trial [12] showed significantly improved pCR rates, with atezolizumab added to nab-paclitaxel, followed by doxorubicin and cyclophosphamide with atezolizumab versus a placebo. Patients in the atezolizumab arm experienced a pCR rate of 58% compared with 41% for those in the control arm, translating to a difference of 17% (95% CI, 6–27%; *p* = 0.0044). However, the study did not reach statistical significance in long-term survival endpoints despite consistent results seen in the PD-L1-positive population: DFS (HR 0.57, 95% CI 0.23–1.43) and OS (HR 0.71, 95% CI 0.26–1.91) numerically favored the atezolizumab containing arm. AEs of special interest occurred in 81% of patients in the atezolizumab arm, including 17% with grade 3/4 events, compared with 61% in the placebo arm, including 13% with grade 3/4 events. Sixteen percent and 10% of patients required steroids in the experimental and control arms, respectively.

In the I-SPY2 trial [20] TNBC cohort, 29 patients received neoadjuvant pembrolizumab with paclitaxel, while 85 patients were treated with paclitaxel alone, followed in all patients by doxorubicin and cyclophosphamide. The estimated pCR rates were 60% and 20%, favoring the addition of pembrolizumab. Adrenal insufficiency signal was seen with pembrolizumab, often with a delayed onset, after doxorubicin and cyclophosphamide treatment completion.

The relationship between pCR and significantly reduced risk of relapse and death compared with patients with residual disease has been previously recognized [4,12,15,22,23]. This thereby made it an important endpoint in early-stage TNBC trials. These observations have been corroborated in a meta-analysis of five randomized trials of 1496 TNBC patients, showing significant improvement in the odds ratio of pCR with the addition of ICI in high-risk TNBC patients [16].

#### 3.1.2. The Role of ICI in Post-Neoadjuvant and Adjuvant Settings

In addition to its role in the neoadjuvant settings, immunotherapy holds promise as a post-neoadjuvant and adjuvant therapy for TNBC. Although standard adjuvant therapies include chemotherapy and radiation, immunotherapy emerged as an adjunct option to conventional therapy. The rationale behind incorporating immunotherapy into the adjuvant setting lies in its potential to target residual disease, eradicate micrometastatic disease in efforts to prevent disease recurrence and achieve durable remission. However, while the combination of chemotherapy and ICI has been studied broadly and is known to be generally safe, the efficacy of this combination in the post-neoadjuvant and adjuvant setting is unknown.

Residual disease after neoadjuvant chemotherapy is a poor prognostic sign in early-stage TNBC [24,25]. Patients who do not achieve pCR have an estimated 5-year event-free survival (EFS) of 57% and overall survival (OS) of 47% compared with 90% EFS and 84% OS, respectively, for patients with early-stage TNBC who demonstrate pCR [23]. 

Post-neoadjuvant therapy could mitigate these outcomes by stratifying adjuvant treatment based on residual disease to identify patients at high risk who may benefit from additional adjuvant therapy. Pembrolizumab has improved disease-free survival (DFS) when added to standard neoadjuvant chemotherapy in patients with high-risk early-stage TNBC. The effect of post-neoadjuvant pembrolizumab seems less robust in recent 5-years data analyses of the KEYNOTE-522 trial as compared to its utility in the neoadjuvant setting and most patients with residual disease have benefited from its use. The largest benefit from pembrolizumab was observed in the RCB-2 group and a striking difference in 3-year EFS between RCB-1 (≃84%) and RCB-3 (≃30%) [26]. Nonetheless, since all patients in the KEYNOTE 522 study received adjuvant immunotherapy, it is not possible to assess if selected patients can forgo this part of the treatment and this is an area of ongoing research to continue to personalize the treatment of patients with early breast cancer [26,27].

Conversely, the addition of Atezolizumab to adjuvant chemotherapy in high-risk early-stage TNBC has proved futility in the ALEXANDRA/Impassion030 phase 3 trial, presented during the 2023 San Antonio Cancer Symposium (SABCS; Abstract GS01-03). This trial was closed following an early interim analysis once a lack of benefit was identified. After monitoring the patients for an average of 32 months, the researchers found no improvement in DFS in those treated with adjuvant atezolizumab as compared with those treated with chemotherapy alone; 12.8% (*n* = 141) vs. 11.4% (*n* = 125), HR 1.11. Results were similar in most patient subgroups, including the PD-L1 positive subgroup.

#### 3.1.3. Immunotherapy in Advanced Triple Negative (mTNBC) Setting

Metastatic TNBC is an aggressive cancer that carries a particularly poor prognosis and limited treatment options. Immunotherapy offers an additional therapy option and a glimpse of hope in this challenging disease. Table 2 summarizes the pivotal trial in this space.

In the advanced and metastatic setting, the KEYNOTE-355 trial showed that pembrolizumab along with chemotherapy produced a statistically significant improvement in survival in patients with PD-L1-positive (CPS > 10) TNBC in first-line setting: mOS: 23 months vs. 16 months; HR: 0.73, 95% CI: 0.55–0.95, *p* = 0.0185; mPFS: 9.7 months vs. 5.6 months; HR: 0.66; 95% CI, 0.50 to 0.88. This study confirmed pembrolizumab with chemotherapy as a first line standard of care in PDL-1 positive mTNBC [5,31]. Similarly, the Impassion130 study [29], utilizing atezolizumab with nab-paclitaxel, demonstrated significantly improved mPFS but failed to do so for mOS due its hierarchal study design, although an exploratory analysis in this subgroup did show a clinically meaningful benefit after median follow-up of 19 months; mOS: 25.4 versus 17.9 months; hazard ratio (HR): 0.67, 95% CI: 0.53–0.86 [29].

In contrast, a similarly designed IMpassion131 trial investigating the combination of atezolizumab and paclitaxel failed to demonstrate a PFS benefit in an untreated PD-L1-positive population. The reasons for this may be multifactorial, attributed at least partly to prior exposure to taxanes in half of the ITT population in early setting and patient heterogeneity [30]. Unlike in the early stage TNBC, immunotherapy seems to derive a meaningful benefit only in select patients with mTNBC, underscoring PDL-1 as the current biomarker of choice for patient selection.

The efficacy of immunotherapy in metastatic TNBC highlights the importance of biomarker profiling in patient selection. PD-L1 expression serves as a predictive biomarker for immunotherapy response, guiding treatment decisions and optimizing outcomes. Certainly, additional markers are needed for better patient selection and the question of partnering agents in the PDL-1 negative mTNBC population remains unanswered. Another important issue is the lack of data as to ICI efficacy in patients with early distant recurrence, within <6 months after neoadjuvant chemoimmunotherapy. These patients with TNBC were excluded from the KEYNOTE 355 and Impassion130 trials. Ongoing research focuses on identifying novel biomarkers and immunotherapy combinations are underway [32].

### 3.2. Immunotherapy in HR Positive, HER-2 Negative Breast Cancer

Most breast cancers are classified as Estrogen Receptor (ER) positive, Progesterone Receptor (PR) positive and HER2 negative. It is defined by ER expression of more than 1%. This subtype of breast cancer is found in approximately 65% of patients and is generally associated with a favorable prognosis [1]. However, it is also associated with low immunogenicity, presumably due to paucity of Tumor-Infiltrating Lymphocytes (TILs), low HLA class I expression, and abundant tumor-associated macrophage, which limit its antitumor immune activity [33]. Furthermore, the prognosis and response to hormonal therapy vary significantly. Lower expression of hormone receptors, higher Ki-67 proliferation index and high grade denote high-risk luminal B-like BC, which carries a worse prognosis. This subtype of breast cancer tends to have higher recurrence rates, posing a challenge and a great need for more efficacious therapies. Recent trials identified this subtype of BC expressing higher levels of TMB, TILs, and PD-L1 supporting the evaluation of ICI in this setting [33,34,35].

Although hormone therapies represent the mainstay of treatment in HR+/HER2− BC, ICI has shown promise as an adjunct to neoadjuvant chemotherapy aimed at improving outcomes in for high-risk patients. Several studies have investigated the role of ICI in combination with chemotherapy and hormonal therapy. Preliminary results suggest that adding immunotherapy to hormone therapy may enhance treatment response and improve outcomes. The rationale behind combining immunotherapy with hormone therapy lies in their complementary mechanisms of action. Although hormone therapy inhibits estrogen signaling, immunotherapy activates immune responses against cancer cells. This synergistic approach aims to enhance treatment efficacy and overcome resistance mechanisms, ultimately improving patient outcomes [35].

Data from few pivotal phase III trials showed a benefit of adding neoadjuvant immunotherapy to chemotherapy in HR+/HER2− BC:

KEYNOTE-756 [36] was a randomized, double-blind Phase 3 trial in 1278 patients with high-risk, early-stage HR+/HER2− breast cancer comparing pembrolizumab in combination with chemotherapy as neoadjuvant treatment, followed by adjuvant treatment with pembrolizumab plus endocrine therapy. The study showed a significantly increased rate of pathological complete response (pCR) in 24.3% with pembrolizumab plus chemotherapy versus 15.6% with placebo plus chemotherapy, representing treatment difference of 8.5%. The pCR benefit was generally consistent across pre-specified subgroups including tumor PD-L1 status (CPS ≥ 1 versus <1), lymph nodes involvement, and estrogen receptor (ER) positivity (≥10% versus 1–9%). However, while many patients enrolled in the study benefited from the addition of pembrolizumab, patients with ER-low disease (ER 1–9%) particularly derived benefit, achieving pCR in 59% of the patients vs. in 30.2% of those who had ER-low disease and received placebo.

In the neoadjuvant phase, grade ≥ 3 treatment-related adverse event (TRAE) rates were 52.5% with pembrolizumab versus 46.4% with placebo.

A similar pCR benefit for 521 women with HR+/HER2− high-risk breast cancer was reported in a second trial, the CheckMate 7FL [37] investigating the role of nivolumab, in the neoadjuvant setting combined with chemotherapy versus placebo plus chemotherapy followed by adjuvant nivolumab or placebo plus endocrine therapy. Reported pCR rates with neoadjuvant nivolumab plus chemotherapy were significantly improved; 24.5% versus 13.8% and the benefit with nivolumab was greater in PD-L1 positive population (CPS > 1) 44.3% versus 20.2% with placebo (OR, 3.11; 95% CI, 1.58–6.11). Although promising, Nivolumab is not currently FDA-approved for breast cancer, and further research is needed to confirm the benefit of this combination therapy. In the neoadjuvant phase, grade 3–4 TRAE incidence was similar across arms (35% versus 32%).

Pembrolizumab also showed benefit in the HR+HER2− arm of the phase II I-SPY2 trial [20], where pembrolizumab plus neoadjuvant chemotherapy improved estimated pCR rates vs. neoadjuvant chemotherapy alone, at 30% vs. 13%, in the cohort of patients with HR+/HER2− breast cancer. This trial is ongoing.

A particular challenge in the HR+/HER2− high-risk and metastatic population is the proven benefit of CDK4/6 inhibition on DFS, and the adverse effects associated with combining these agents with ICI. Despite the known synergistic activity of ICI and CDK4/6i [38,39], this combination resulted in excess toxicity, specifically high rates of interstitial lung disease and liver injury, resulting in discontinuation of enrollment [40,41].

As in TNBC, careful patient selection is of utmost importance to determine which patients with HR+/HER2-BC derive the most benefit, while minimizing adverse effects, considering deaths were reported in patients with early-stage disease treated with ICI. 

### 3.3. Immunotherapy in HER2+ Breast Cancer

HER2+ is an aggressive subtype of breast cancer that accounts for 20–25% of all breast cancers. It is associated with an increased risk of recurrent disease and carries a worse prognosis as compared to its HR-positive HER2-negative counterpart [1]. This subtype of BC is characterized by higher PDL-1 expression and Tumor-Infiltrating lymphocytes (TILs) infiltration. Although the treatment landscape for HER2-positive breast cancer has shifted with the advent of anti-HER therapies, it continues to pose significant challenges as an aggressive and often fatal disease. The utilization of immune checkpoint inhibitors in HER2-positive breast cancer is underpinned by compelling biological and preclinical evidence. There is preclinical data to indicate ICI’s role in overcoming the resistance to seen to trastuzumab [41] and this combination has been evaluated in several clinical trials.

Although no checkpoint inhibitors are currently FDA-approved in this space, there is a strong rationale for combining HER2-targeted therapies with ICI to increase efficacy in breast cancer, particularly in the early-stage setting, where the immune system has not been weakened by heavy pretreatment. Few trials have shown the benefit of immunotherapy in combination with anti-HER2 directed therapy, though more robust research is needed for regulatory approval incorporation into clinical guidelines.

PANACEA was a single-arm, phase 1b-2 proof-of-concept trial that demonstrated that combining trastuzumab with pembrolizumab, had some benefit in 58 patients with trastuzumab-resistant, heavily pretreated, HER2-positive advanced BC. The overall response rate was 15% and was limited to PDL-1-positive tumors. A significant difference of 65% versus 12% in the 12-month OS rate was observed between PD-L1+ and PD-L1− tumors, accordingly; however, several limitations including a small sample size and the absence of a control arm make it difficult to define the prognostic or predictive value of PD-L1 expression [42].

The Keyriched-1 trial was the first exploration of a chemotherapy-free neoadjuvant treatment approach along with immunotherapy. It assessed the combination of pembrolizumab with HER2-targeted therapies, trastuzumab, and pertuzumab, for HER2-enriched early breast cancer. In the primary analysis, this triplet regimen exhibited pCR rate of 46% (95% CI: 0.31–0.62), with higher pCR rates in the HR−/HER2+ tumors 58.5% compared with HR+/HER2+ tumors 38.5% [43].

When referring to the PDL-1 inhibitor atezolizumab in this space, the results are mixed. Although Neo-PATH trial evaluated atezolizumab combined with HER2-targeted therapies and chemotherapy in the neoadjuvant setting resulting in a pCR rate of 61% in the intention-to-treat population [44], the IMpassion050 has failed to demonstrate pCR improvement as compared with placebo [45].

Despite mostly positive results, the trials in the HER2 + BC, as in the HR+/HER2− space, only show a pCR and response rates benefits rather than efficacy-driven endpoints, which adds to the growing body of evidence linking ICI with pCR. Nonetheless, survival-based endpoints are needed to underpin the effectiveness of ICI in HER2 + BC.

## 4. The Role of Biomarkers

Certain biomarkers can predict the response to ICI across different BC subtypes [9,13,14]. To date, positive PDL-1 CPS is an established biomarker needed per FDA approval to utilize pembrolizumab in first line setting mTNBC. It is the biomarker of choice in mTNBC, predictive of pCR, and prognostic of superior outcomes in the first-line setting [5]. Notably, early TNBC tend to respond to ICI regardless of PDL-1 CPS, though PD-L1 positivity predicted higher rates of pCR [6,12]. Attempts to develop ICI in an unselected mTNBC population resulted in no added benefit, and the use of ICI in the subsequent lines showed inferior overall response. Other biomarkers such as Tumor Mutation Burden (TMB) and tumor-infiltrating lymphocytes (TILs) hold predictive values for ICI response and serve as independent prognostic factors to guide selection for immunotherapy [34].

### 4.1. Tumor Mutation Burden (TMB)

Hypermutated TMB is defined as ≥10 mut/Mb. Though a marker for pembrolizumab eligibility [46], it is not commonly elevated in breast cancer as compared to other malignancies. One study analyzed data from 3966 breast tumors included in eight cohorts reporting approximately 5% of all breast cancers had a TMB ≥ 10 mut/Mb, and metastatic tumors had a greater prevalence of high TMB than primary tumors (8.4% versus 2.9%) [47]. Although TNBC has the highest median TMB of all subtypes of BC, the frequency of hypermutated tumors (≥10 mut/Mb) is similar among different subtypes. Metastatic invasive lobular carcinoma was the highest hypermutated histology type, more so than invasive ducal carcinoma (17.0% versus 7.8%) [48,49]. High TMB is associated with longer PFS and OS among patients with mTNBC following treatment with ICI, independent of clinical factors and PD-L1 status. This association was not seen in mTNBC patients treated with chemotherapy alone, underlying the link between hypermutated TMB and survival benefit when treated with ICI [50].

Interestingly, in the neoadjuvant setting, TMB demonstrated to be a marker for pCR prediction in multivariate analysis, independently from PDL-1 CPS [51]. This association was not seen in the HR+/HER2− mBC setting and more data are needed to establish a TMB cutoff to be used in breast cancer.

### 4.2. Tumor-Infiltrating Lymphocytes (TILs)

The discovery of TILs in aggressive subsets of breast cancers, such as TNBC and HER2+ established it as another biomarker that holds prognostic and predictive value for patients’ outcomes. Intratumor TILs decrease from early- to late-stage disease and with increasing tumor burden; perhaps at least partially explaining this difference. Higher TILs are prognostic for improved outcomes in early-stage disease regardless of treatment: High TILs presence in the tumor is associated with improved 5-year overall survival (OS; 74.3% in the high TIL group vs. 52.0% in the low TIL group), higher rates of pCR; 40–50% in the high TIL group after chemotherapy, and improved DFS particularly in TNBC [52,53]. The NeoPACT phase II study evaluated combination anthracycline free chemotherapy with pembrolizumab, and reported superior pCR outcomes in patients with high TILs and other immune response markers [54]. Recent data suggest the linear connection between stromal Tumor-infiltrating lymphocytes (sTILs) and outcomes in patients with eBC TNBC whereby patients with sTILS of 50% or over and T1c tumors had 10-year survival of 95% without chemotherapy, increasing to 98% when sTILS were 75% or greater. These findings further establish sTILS as a biomarker for de-escalation strategies [55,56].

Research is underway to identify signatures of tumor immune responsiveness to better identify patients with immunotherapy-targetable BC with examples including a 27-gene TME assay [57], enhanced immune (Immune+) [58], and the ImPRINT assay [59,60,61].

## 5. Combinations: ICI with Antibody Drug Conjugates (ADC)

Regarding the combination of ICI and ADC, the randomized phase II KATE2 trial examined the addition of atezolizumab to trastuzumab emtansine (T-DM1) in patients with HER2-positive mBC who had progressed on trastuzumab and taxanes. Although it did not meet its primary endpoint of progression-free survival (PFS) in the intention-to-treat (ITT) population, a notable difference emerged in the PD-L1-positive subgroup (84 out of 202 patients, 42%), with a median PFS of 8.5 months compared to 4.1 months (hazard ratio: 0.60, 95% CI: 0.32–1.11); however, no significant differences were observed in overall survival (OS), which was a secondary endpoint [62]. The ongoing phase III KATE3 clinical trial (NCT04740918) is investigating this combination exclusively in PD-L1-positive patients, with both PFS and OS as co-primary endpoints. Additionally, the potential benefit of adding atezolizumab to T-DM1 is being explored in the post-neoadjuvant setting among patients with residual disease following neoadjuvant therapy (ASTEFANIA trial, NCT04873362).

The combination of pembrolizumab and sacituzumab govitecan-hziy, a TROP2-directed ADC, has shown a synergetic effect whereby preclinical data suggest that SG potentiates the activity of ICIs. This combination was trialed in a phase II study (NCT04448886) in HR+, HER2− mBC, patients unselected by PD-L1 status. The study resulted in a numerical but not statistically significant improvement in median PFS of 8.12 months (95% CI, 4.51–11.12) vs. 6.22 months (95% CI, 3.85–8.68) in the antibody-drug conjugate (ADC) alone, as was presented during the 2024 ASCO Annual Meeting. 

The addition of sacituzumab to a pembrolizumab-based adjuvant therapy in early-stage high-risk TNBC patients with residual disease after neoadjuvant therapy and surgery will be evaluated in a phase III ASCENT-05/OptimICE-RD study (NCT05633654) that is currently ongoing.

Another promising ADC, T-DXd, has demonstrated remarkable antitumor activity in both HER2-positive and HER2-low breast cancer [63]. The safety profile of combining T-DXd with nivolumab was evaluated in a phase Ib trial involving 48 patients (32 HER2-positive and 16 HER2-low), with 50% experiencing prohibitive treatment-related adverse events of grade 3 or higher, leading to treatment discontinuation in 37% of patients (25% attributed to T-DXd, 21% to nivolumab) [64]. The ongoing phase Ib/II DESTINY-Breast 07 trial (NCT04538742) investigates the combination of T-Dxd with durvalumab and paclitaxel.

## 6. Immunotherapy beyond ICI

Regarding therapies other than ICIs, chimeric antigen receptor (CAR)-T cells have emerged as a promising immunotherapeutic strategy in TNBC: this approach combines the antigen specificity of an antibody with the effector function of T cells. It is under investigation in several phase I/II clinical trials. Although numerous antigens have been identified as potential targets (e.g., Trop2, GD2, ROR1, MUC1, EpCAM), the ideal target should represent the most relevant obstacle, thereby minimizing on-target/off-tumor toxicities and reducing tumor escape via antigen loss and intrinsic heterogeneity [65,66].

Several emerging immune checkpoint targets are under investigation, including lymphocyte activation gene 3 (LAG-3) [67]. Research on LAG-3 in breast cancer has revealed its potential as a prognostic factor and therapeutic target. Studies have shown that LAG-3 expression is associated with improved survival in HR negative breast cancer, particularly when co-expressed with CD8+ TILs [68]. It has also been linked to the malignancy of breast cancer and may synergize with other immune checkpoints, such as CTLA4 and PD1/PDL1 [69]. In TNBC, LAG-3 expression is associated with improved overall and recurrence-free survival, and its co-expression with PD-L1 suggests potential for combination treatment [70]. However, the impact of LAG-3 expression on metastasis-free survival in early breast cancer is less clear, with some studies suggesting a favorable outcome [71] and others finding no significant impact [72].

Despite years of research efforts, cancer vaccines have shown confounding results. Early I/II trials targeting HER peptide showed no significant clinical benefit [73,74] and a subsequent phase III trial involving E75 vaccine of patients, including TNBC with node-positive HER2-low expressing breast tumors was stopped early when an interim analysis failed to demonstrate a significant difference in DFS between E75 vaccinated and placebo vaccinated patients [75]. However, a meta-analysis of 24 clinical studies of 1704 vaccinated patients and 1248 control subjects found that E75 vaccination caused significant improvement in disease recurrence rate and DFS but no significant difference in OS [76]. In the TNBC space, a dendritic cell vaccine targeting HER2 and HER3 has been used to treat patients with brain metastases [77]. An ongoing study that targets multiple commonly expressed TNBC antigens is the PVX-410, which is being evaluated in early-phase trials [78,79]. A novel approach to primary prevention of TNBC was trailed in high-risk, BRCA1-carrying women, whereby vaccination against the human lactation protein, α-lactalbumin, may provide a safe and effective strategy for primary prevention harnessing α-lactalbumin expression exclusively in the breast and only during late pregnancy and lactation but is expressed in >70% of TNBCs [80]. This vaccine is also being trialed at an adjuvant setting for early-stage TNBC and is in ongoing. Other vaccines targeting tumor-associated carbohydrate antigens, P10s-PADRE and non-protein hexasaccharide with a ceramide, the Globo H glycosphingolipid antigen, has reached phase 3 clinical trial status in patients with Globo H+ TNBC tumors [81]. Oncolytic viruses have shown promise in the treatment of breast cancer, particularly in advanced and metastatic stages. These viruses, including herpes simplex virus, adenovirus, vaccinia virus, measles virus, and reovirus, have demonstrated safety and efficacy in early-phase clinical trials. The clinical utility of oncolytic virus therapy in breast cancer remains to be determined and to date, no breakthrough study has emerged with this technique. Trials evaluating the utility of oncolytic viruses in direct tumor killing in TNBC are underway [82,83,84].

## 7. Future Directions and Challenges

Though slower integration in breast cancer, the development of cancer immunotherapy has transformed the current treatment landscape of TNBC and will hopefully expand beyond the TNBC subtype. Fortunately, the number of clinical trials evaluating multiple immunotherapeutic strategies is increasing across all BC subtypes. There is great anticipation for real-world data that would come out from the regimens utilizing ICI with chemotherapy as these begin to be incorporated into the standard practice of patients, to provide a better understanding of the risks and benefits of this approach. Importantly, there is a need for real-world data to capture the outcomes of underrepresented populations in landmark clinical trial populations. 

Despite the groundbreaking progress achieved with immunotherapy for patients with breast cancer, there are lingering challenges. Biomarker identification remains crucial for optimal patient selection and improving the risk-benefit ratio. Recent data establish sTILS as a biomarker for de-escalation strategies in TNBC and potentially obviating the need for chemotherapy altogether. As outcomes improve, we face the challenge of long-term follow-up that is needed to capture survival benefits. To this end, the development of novel surrogate endpoints and imaging modalities to measure the immune response could refine the assessment of tumor response and predict the benefit of a given therapy. Another challenge is to define adequate response criteria, as the pattern of responses to ICI may be different from that of chemotherapeutic agents. Immune Response Evaluation Criteria in Solid Tumors (iRECIST) to better capture the benefit of immunotherapy have been developed, but most trials are still using the conventional RECIST [85].

Future research endeavors focusing on refining immunotherapy delivery, elucidating resistance mechanisms, and exploring novel combination therapies are underway, offering a promising avenue toward more effective and targeted treatments for breast cancer patients.

## 8. Summary and Discussion

Cancer immunotherapy represents one of the most significant advances in oncology in recent years. In particular, the utilization of Immune Checkpoint Inhibitors (ICIs) demonstrated impressive anti-tumor activity and a durable clinical benefit. In triple-negative breast cancer (TNBC), the incorporation of pembrolizumab along with chemotherapy in high-risk eBC and in select patients with advanced TNBC has become the current standard of care and has transformed the landscape of this aggressive entity. Immune checkpoint inhibitors have also demonstrated superior outcomes in the preoperative setting, aiding in achieving pCR rates in the HR+/HER− eBC but also independently, showing superior outcomes amongst patients achieving pCR when compared to chemotherapy alone. This suggests a lingering effect on the tumor microenvironment, that translates into reduced risk of recurrence. Clinical trials have shown that ICIs are most effective when used up front, as part of neoadjuvant therapy, especially when the immune microenvironment is more favorable, and evasion is limited.

Extensive research is underway to better understand the immune landscape of other breast cancer subtypes, where ICIs could potentially offer similar benefits as seen in TNBC. Several ongoing clinical trials are exploring ICIs to enhance clinical outcomes. Among these, the combination of ICIs with antibody-drug conjugates (ADCs) appears most promising and might replace the use of naked chemotherapy in combination with ICI in the future. Special attention on identifying and mitigating the added toxicities with these combinations, while maintaining a favorable risk-benefit ratio is of utmost importance. The success of treatment escalation or de-escalation strategies hinges on identifying and validating biomarkers for response and resistance. There is a need to refine the use of existing biomarkers for patient selection and to identify novel biomarkers with both predictive and prognostic value for ICI utilization.

Novel immunotherapeutic agents, such as CAR-T cell therapy, oncolytic viruses, and tumor vaccines, represent innovative approaches for advanced and early-stage breast cancer, holding promise for improved outcomes for breast cancer patients.

## Figures and Tables

**Table 1 ijms-25-07517-t001:** Immunotherapy in early-stage TNBC.

	KEYNOTE 522 [4,6]	IMPASSION 031 [12]	I-SPY2 [20]
Phase	III	III	II
No. of patients	1174	333	114
Study design	pembrolizumab/placebo 200 mg q3w × C8 + carboplatin AUC5 q3w or AUC1.5 q1w + paclitaxel 80 mg/m^2^ q1w × C4, followed by doxorubicin 60 mg/m^2^ q3w × C4 cycles or Epirubicin 90 mg/m^2^ q3w × C4 + cyclophosphamide 600 mg/m^2^ q3w × C4 followed by adjuvant pembrolizumab/placebo × C9.	atezolizumab/placebo 840 mg q2w + nab-paclitaxel 125 mg/m^2^ × C12, followed by ddAC × C4, followed by adjuvant atezolizumab 1200 mg q3w × C11 vs. capecitabine for non pCR or follow up.	Paclitaxel 80 mg/m^2^ q1w × C12 +/− pembrolizumab 200 mg q3w × C4, followed by AC q3w × C4.
Primary endpoint(s)	pCR, EFS	pCR in ITT and PDL-1 +	pCR in ITT
% pCR (ICI vs. placebo)	65% vs. 51% *p* < 0.001	pCR: 58% vs. 41% (ITT); 69% vs. 49% (PDL-1 +) *p* = 0.004	pCR: 60% vs. 22%
EFS/DFS/OS (ICI vs. placebo)	5yEFS 81.3% vs. 72.3%	PDL1+: DFS (HR 0.57, 95% CI 0.23–1.43); OS (HR 0.71, 95% CI 0.26–1.91)	N/A. Not powered for statistical significance.
Outcome summary	The addition of pembrolizumab to NAC reduced the risk for recurrence, progression, complications, or death by 37% (HR, 0.63; 95% CI, 0.49–0.81).	atezolizumab increased pCR rate by 17%. No statistical significance in long-term survival endpoints despite consistent results in the PD-L1-positive population, numerically favored the atezolizumab.	pembrolizumab increased pCR rates by 38%.
Immure Related Adverse Effects Grade > 3	82% vs. 79%	23% vs. 16%	9% vs. 7%

Footnote: dd, dose-dense; AC, doxorubicin + cyclophosphamide; AUC, area under curve; DFS, disease-free survival; EFS, event-free survival; ITT, intention to treat; OS, overall survival; PFS, progression-free survival; pCR, pathological complete response; PDL1, programmed death-ligand 1; +, positive; TNBC, (early/metastatic) triple-negative breast cancer; W, week.

**Table 2 ijms-25-07517-t002:** Immunotherapy in advanced/metastatic TNBC.

	IMpassion 130 [28,29]	IMpassion 131 [30]	KEYNOTE-355 [5]
Phase	III	III	III
No. of patients	943	902	847
Study design	Atezolizumab/placebo 840 mg D1,15 q4w + nab-paclitaxel 100 mg/m^2^ D1,8,15 q4w	Atezolizumab/placebo 840 mg d1,15 + paclitaxel 90 mg/m^2^ d1,8,15) q4w	Pembrolizumab/placebo 200 mg q3w + CT (nab paclitaxel; paclitaxel; or gemcitabine + carboplatin)
Primary endpoint(s)	PFS (ITT; PD-L1 ≥ 1%); OS (ITT; PDL1 ≥ 1%)	PFS in ITT and PD-L1+	PFS, OS
ORR (ITT; PDL-1+)	ORR 56% vs. 46% (ITT); 59% vs. 43% (PDL1+); mDOR: 7.4 vs. 5.6 mo (ITT); 8.5 vs. 5.5 m (PD-L1+)	ORR: 53.6% vs. 47.5%	ORR: 53% (PDL-1+)
PFS (ITT; PDL-1+)	7.2 vs. 5.5 mo HR 0.79 (95% CI 0.69–0.91, *p* = 0.002); 7.5 vs. 5.3 mo HR 0.63 (95% CI 0.5–0.8 *p* < 0.0001)	mPFS: 5.7 vs. 5.6 mo; HR 0.86 (95%CI 0.7–1.05); 5.7 mo vs. 6 mo HR 0.82 (95% CI 006–1.12, *p* = 0.2)	7.5 vs. 5.6 mo HR 0.82 (95% CI 0.69–0.97); 9.7 vs. 5.6 mo HR 0.65 (95% CI 0.49–0.86 *p* = 0.0012)
OS (ITT; PDL-1+)	21.0 vs. 18.7 mo HR 0.87 (95% CI 0.75–1.02, *p* = 0.077); 25.4 vs. 17.9 mo HR 0.67 (95% CI 0.53–0.86).	mOS: 22.8 vs. 19.8 mo HR 1.12 (95% CI 0.88–1.43); 28.3 vs. 22.1 mo HR 1.11 (95% CI 0.76–1.64)	17.2 vs. 15.5 mo HR 0.89 (95% CI 0.76–1.05); 23.0 vs. 16.1 mo HR 0.73 (95% CI 0.55–0.95, *p* = 0.0093).
Outcome summary	Atezolizumab with nab-paclitaxel demonstrated significantly improved mPFS but failed to do so for mOS due its hierarchal study design. An exploratory analysis in PD-L1+ * subgroup did show a clinically meaningful mOs benefit at median follow-up of 19 months.	The combination of atezolizumab and paclitaxel failed to demonstrate survival benefits.	Pembrolizumab along with chemotherapy produced statistically significant improvement in survival in patients with PD-L1+ ** TNBC in first-line setting.
Immure Related Adverse Effects Grade > 3	7.5% vs. 4.3%	62% vs. 53% (all grades)	5.3% vs. 0%

Footnote: * For atezolizumab, PD-L1 is defined by <1% expression on immune cells as determined by Ventana SP142 IHC assay. ** For pembrolizumab, PD-L1 is defined by CPS < 10 as determined by Dako 22C3 IHC assay. CPS is calculated by adding all PD-L1–positive cells (lymphocytes, macrophages, and tumor) divided by the total number of viable tumor cells.

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
