# Peer review of "Immunotherapy in Breast Cancer"

_ijms, 2024, doi:10.3390/ijms25147517_

Round 1

Reviewer 1 Report

Comments and Suggestions for Authors

Thank you for inviting me to evaluate the review article titled “Immunotherapy in Breast Cancer”. This study thoroughly presented the current landscape of immunotherapy, particularly checkpoint inhibitors, in different types of breast cancer. The authors describe the review paper rationale, I have some minor comments as below.

1. Expand the introduction to provide a more detailed context for this review paper. Highlight the significance of immunotherapy in breast cancer and the gap this review aims to fill.

2. While the paper reviews current immunotherapy approaches, it would be helpful to include more detailed descriptions of the methods used in the studies reviewed, like how the author selected papers and clinic trials.

3. Ensure all tables are clearly labeled and referenced in the text, please also add all the references in table 1 and 2 for each trial.

4. It would be better if the author could mention any promising trials or studies currently underway in this review paper.

5. Reinforce the need for future research and specify the types of studies that would be most beneficial.

Author Response

Thank you for your review. Your comments have been implemented to the best of our abilities. 

Thank you for inviting me to evaluate the review article titled “Immunotherapy in Breast Cancer”. This study thoroughly presented the current landscape of immunotherapy, particularly checkpoint inhibitors, in different types of breast cancer. The authors describe the review paper rationale, I have some minor comments as below.

  1. Expand the introduction to provide a more detailed context for this review paper. Highlight the significance of immunotherapy in breast cancer and the gap this review aims to fill.
    Added in introduction.

  2. While the paper reviews current immunotherapy approaches, it would be helpful to include more detailed descriptions of the methods used in the studies reviewed, like how the author selected papers and clinic trials.
    Added in the introduction section.

  3. Ensure all tables are clearly labeled and referenced in the text, please also add all the references in table 1 and 2 for each trial.
    Done

  4. It would be better if the author could mention any promising trials or studies currently underway in this review paper.

  5. Reinforce the need for future research and specify the types of studies that would be most beneficial.
    Added.

Reviewer 2 Report

Comments and Suggestions for Authors

This manuscript titled as “Immunotherapy in Breast Cancer”, provides insights into the role of immunotherapy in breast cancer, integrating literature review with clinical trial data. The identification of immunotherapy as a viable treatment option across various breast cancer subtypes is particularly promising, and further exploration of its mechanisms and long-term significance will be valuable for developing effective treatments for breast cancer. However, novel is very limited in this manuscript. In addition, apparent flaws can be also observed in this manuscript. Thus, major revision is recommended. 

1.     This manuscript titled as “Immunotherapy in Breast Cancer”. However, authors only summarized and discussed the treatment of ICIs. How about CAR-T, cancer vaccines, Cytokine Therapy, Oncolytic Virus Therapy, etc?  Substantial work has been missed in this article, apparently. 

2.     As for the ICIs treatment in breast cancer, how are the agents for inhibiting other immune checkpoint LAG-3, TIGIT,TIM-3, etc?

3.      The introduction should emphasize the rationale for focusing on immunotherapy within breast cancer, highlighting the challenges and gaps in current treatment modalities.

4. A clear description of how studies were selected for inclusion, the criteria used for evaluating their quality, and the methods for data extraction and synthesis, is recommended. 

5.A more detailed explanation of the significance of immunotherapy in the context of breast cancer, is recommended.

6. More comprehensive content is required in the section of “Future Directions and Challenges”.

Author Response

Thank you for your review. Your comments have been implemented.

This manuscript titled as “Immunotherapy in Breast Cancer”, provides insights into the role of immunotherapy in breast cancer, integrating literature review with clinical trial data. The identification of immunotherapy as a viable treatment option across various breast cancer subtypes is particularly promising, and further exploration of its mechanisms and long-term significance will be valuable for developing effective treatments for breast cancer. However, novel is very limited in this manuscript. In addition, apparent flaws can be also observed in this manuscript. Thus, major revision is recommended. 

  1. This manuscript titled as “Immunotherapy in Breast Cancer”. However, authors only summarized and discussed the treatment of ICIs. How about CAR-T, cancer vaccines, Cytokine Therapy, Oncolytic Virus Therapy, etc?  Substantial work has been missed in this article, apparently. 
    These agents are mentioned in the Immunotherapy Beyond ICI  section. We have expanded this section further.

  2. As for the ICIs treatment in breast cancer, how are the agents for inhibiting other immune checkpoint LAG-3, TIGIT,TIM-3, etc?
    This review did not intend to go into deeper resolution of these agents. However, they were mentioned in section 6 and we expanded on it further.

  3. The introduction should emphasize the rationale. for focusing on immunotherapy within breast cancer, highlighting the challenges and gaps in current treatment modalities.
    Additional content added.

  4. A clear description of how studies were selected for inclusion, the criteria used for evaluating their quality, and the methods for data extraction and synthesis, is recommended. 
    Added in introduction.

  5. A more detailed explanation of the significance of immunotherapy in the context of breast cancer, is recommended.
    Added in section 8.

  6. More comprehensive content is required in the section of “Future Directions and Challenges”.
    Added

Round 2

Reviewer 1 Report

Comments and Suggestions for Authors

The author had already addressed most of my concerns.

Reviewer 2 Report

Comments and Suggestions for Authors

The manuscript "Immunotherapy in Breast Cancer" provides a review of the current landscape of immunotherapy, particularly immune checkpoint inhibitors (ICIs), in the treatment of breast cancer. The manuscript covers various subtypes of breast cancer, discussing the efficacy of ICIs in each subtype, current clinical trials, and future directions. The discussion is supported by relevant literature and data from pivotal trials, making it a valuable contribution to the field. After revision, acceptance is recommended at present stage.